# Potential for Food Self-Sufficiency Improvements through Indoor and Vertical Farming in the Gulf Cooperation Council: Challenges and Opportunities from the Case of Kuwait

**Meshal J. Abdullah \*, Zhengyang Zhang** 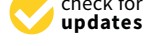 **and Kazuyo Matsubae**

Graduate School of Environmental Studies, Tohoku University, Sendai 980-8572, Japan;
zhengyang.zhang.a8@tohoku.ac.jp (Z.Z.); kazuyo.matsubae.a2@tohoku.ac.jp (K.M.)
\* Correspondence: de.9k2mx@gmail.com

**Abstract:** The countries of the Gulf Cooperation Council (GCC) are considered food secure due to their ability to import sufficient food to meet their populations' demand, despite considerable environmental limitations to conventional agriculture. However, over-reliance on externally produced food leaves these countries vulnerable to food shortages during crises that disrupt international production and shipping. Advanced Controlled Environment Agriculture technology has the potential to improve food self-sufficiency by multiplying vegetable crop yields while optimizing efficiency of agricultural inputs and minimizing land requirements. This paper demonstrates how approximately 15 km$^2$ of indoor farms or less than 0.1 km$^2$ of vertical farms could reduce or eliminate the need to import six important vegetable crops in the State of Kuwait. If properly contextualized and supported by clear legislation and well-managed regulatory bodies, indoor agriculture initiatives may provide a pathway for GCC countries to reduce their dependence on imported foods and increase resilience to food supply disruption during disasters or conflict. This case study contextualizes the need for improved food self-sufficiency in light of vulnerabilities from regional and global threats, illuminates unique challenges faced by GCC countries considering adoption of the proposed technologies, and summarizes opportunities inherent in the current legal and policy framework.

**Keywords:** controlled environment agriculture (CEA); soilless cultivation; food security; urban agriculture; food sovereignty

## 1. Introduction

Despite considerable limitations to conventional agriculture, the countries of the Gulf Cooperation Council (GCC), including Bahrain, Kuwait, Oman, Qatar, Saudi Arabia, and the United Arab Emirates, rank in the top 40% of the 2019 Global Food Security Index [1]. To compensate for harsh climatic conditions, extreme water stress, and minimal arable land, these oil rich nations have forged a robust network of international trade and foreign agricultural investments to ensure the availability of high quality, nutritious food for their populations [2–6]. However, the COVID-19 pandemic and the resulting disruption of transportation and trade brought into sharp focus the vulnerability of this strategy and highlighted the distinction between food security and food self-sufficiency. Without finding innovative new methods to produce more food within their own borders, GCC countries will remain vulnerable to food security risks in the face of global or regional shocks.

Controlled Environment Agriculture (CEA) in the form of indoor and vertical farms has the potential to enhance the yield of conventionally grown crops through the use of hydroponic, aeroponic, or aquaponic technology in seasonally independent, enclosed systems. Bypassing the resource limitations of the external environment, these systems can produce high quality, safe and nutritious crops that may contribute to improved availability, access, utilization, and sustainability of food in GCC countries [7]. If properly contextualized and supported by clear legislation and well-managed regulatory bodies,

indoor agriculture initiatives may provide a pathway for GCC countries to reduce their dependence on imported foods and increase resilience to food supply disruption during disasters or conflict, functioning as a safety net in times of economic crisis.

The objective of this paper is to examine the potential for indoor and vertical farming to alleviate some of the pressing food security vulnerabilities in GCC countries through a case study of Kuwait. Actual cultivated area and crop yield data acquired from the State of Kuwait Public Authority for Agriculture Affairs and Fish Resources (PAAFR) was used to calculate the estimated area of indoor or vertical farms that could achieve nearly 100% self-sufficiency for six important vegetable crops (tomato, potato, green pepper, carrot, lettuce, and cabbage). The paper also brings to light key challenges to implementing indoor and vertical farming in the GCC region, such as complications arising from coupled water and energy production, then explores how these countries are well-positioned to overcome these challenges. Finally, this paper details how indoor agriculture initiatives align with existing national development policies and international commitments, and recommends increased institutional and cross-sector coordination to maximize the potential for environmental, economic, and social benefits.

### 1.1. GCC and Kuwait Context

The countries of the GCC cover nearly 80% of the Arabian Peninsula, sharing similar climatic conditions and geographical features, as well as comparable social and economic contexts. This makes some general comparisons possible among countries of the region. A case study of the State of Kuwait may therefore be used to draw broad conclusions about the potential impact of advanced CEA in the region.

Overlooking the Persian Gulf to the east, Kuwait shares a 195 km border with the Republic of Iraq in the north and a 495 km border with the Kingdom of Saudi Arabia in the south [5]. In total, Kuwait's land mass covers 17,818 km$^2$ with an altitude ranging from sea level up to 300 m [8,9]. Kuwait is classified as a hyper-arid desert with no internal renewable water resources and long-term average annual precipitation of 121 mm [8]. Average high temperature in summer is 45 °C, accompanied by fluctuating humidity and frequent dust storms with wind speeds that can reach up to 150 km per hour [5].

### 1.2. Current State of Agricultural Production

Kuwait is considered one of the least agricultural countries in the world, with less than 9% of its total land area considered arable and less than 1.1% of its work force associated with agricultural activities [2,10]. Of the 1539 km$^2$ of agricultural land, only 121 km$^2$ is under cultivation for vegetable crops [10]. 85% of crops are grown in open field conditions, with 15% grown in greenhouses or other protective covers. However, 100% are fully irrigated [8,10]. Although only a small segment of the economy, the agriculture sector accounted for over 62% of total water withdrawals in the country in 2017, a figure which has steadily grown over the past two decades [8]. Intensive irrigation increases soil salinity, reducing land productivity and requires excess energy for pumping and desalination, estimated between 5–12% of annual electricity consumption in the region [2,4,11,12]. Extreme water scarcity and limited arable land make the challenge of a sustainable and economically viable agricultural industry that can attain food self-sufficiency very difficult.

### 1.3. Consumption Patterns

Crude oil production forms the basis of Kuwait's economy, accounting for nearly half the GDP, 95% of exports, and close to 90% of government revenue [4,5]. With its successful oil industry, Kuwait's population has more than doubled in the past two decades, from 2,045,123 in 2000 to 4,270,563 in 2020 [13]. This growth is attributable to an influx of expatriates which make up 70% of the population, with Kuwaiti citizens numbering under 1.5 million. Uniquely, Kuwait has a 100% urban population [8]. This burgeoning, mixed and metropolitan population demographic, along with Kuwait's status among the richest

countries in the world, has resulted in increased need for food quantity and for food of higher quality and greater diversity than ever before.

### 1.3.1. Food Consumption

The country's agricultural reports show that most of its arable land is used as pasture for livestock, yet only 15% of the demand for red meat is being produced domestically [10]. Only 9% of demand for fish and 19% of demand for shrimp are being met domestically, while less than 0.05% of the need for the staple grains of wheat, barley, and corn is satisfied by local production. However, for fresh vegetables, Kuwait has reached 51% self-sufficiency overall, though the figures vary significantly amongst specific crops. Of the ten most-consumed crops, only four (tomato, cucumber, eggplant, and pumpkin) had domestic production rates that satisfied greater than half the demand. Some important crops, such as onions, garlic, and carrots had extremely low production rates compared to demand, achieving self-sufficiency rates of only 7%, 3% and 2% respectively [10].

### 1.3.2. Water Consumption

With no renewable sources of freshwater, Kuwait is water-scarce. Excess groundwater withdrawal—at least 12 times greater than annual recharge—places Kuwait amongst the top 10 countries with extremely high baseline water stress [3,4,14]. Desalination of seawater is used to compensate for lack of surface and groundwater resources [15]. Nevertheless, water consumption has increased in tandem with food consumption, with the desalination of 1.89 billion liters and consumption of 1.87 billion liters during 2020, the highest ever for the country [16,17]. Throughout the GCC, water consumption is comparable and sometimes greater than that of countries with greater water resources [12]. With 62% of this water being used in the agricultural sector, a critical consideration when undertaking any initiative to improve food self-sufficiency must be to seek out technologies for maximizing food production without compounding the region's water burden [8].

### *1.4. Threats to Regional Food Security*

With environmental limitations on domestic food production, Kuwait relies heavily on food imports, stockpiling, and foreign agricultural investment to satisfy its consumption demands [4,6]. Being an oil-rich country allows it to financially support its food market, making it resistant to global food price fluctuations [3]. This strategy has earned it a rank amongst the top food secure countries in the Middle East region [1]. Yet, Kuwait is not immune to global or regional shocks and the outsourcing of the majority of its food production can be a significant vulnerability.

### 1.4.1. Regional Political Instability

Given Kuwait's sensitive geographical position situated between countries with different and often opposing political and ideological viewpoints, threats and actual disruption of import corridors, including the critical Strait of Hormuz, is a recurring concern requiring the government to adapt to the fluid and sometimes unpredictable nature of intra-regional affairs [3,4,18,19]. This is in addition to the fact that Kuwait's largest agricultural lands lies in vulnerable locations, directly on the country's northern border with Iraq and on its southern border with Saudi Arabia, where a neutral zone further complicates some aspects of the administration and security of Kuwait's most significant agricultural holdings [20]. Because of its unique position and political weight within the region, Kuwait often finds itself soothing tensions and supporting its regional neighbors in overcoming some of their political and even sustainability challenges, such as the recent blockade of Qatar and the ongoing conflict in Yemen [9,21–25].

### 1.4.2. Inter-Regional Relations

Relationships with exporting countries often go beyond simple trade, as Kuwait and other GCC countries leverage their wealth to engage in new strategies such as foreign land

acquisitions and investments in private agricultural crop production abroad to secure their claim to the necessary quantity of food and to diversify the sources for these imports [11,26]. Nevertheless, these strategies still rely on outside sources and depend on international political and economic ties that are directly influenced by the Gulf region's stability, along with the need to consider the issues of responsible and ethical investment in other countries' food, economic infrastructure and shared use of resources [6].

The share of imports coming from various regional groups is diffuse. Although the largest share of food imports originates from within Kuwait's own Middle East and North Africa region (including other GCC countries), this still only accounts for about 30% of total food imports [27]. Countries in Europe and Central Asia provide a similar amount of total food imports (27%), with South Asia (13.5%) and East Asia and the Pacific (12.5%) collectively contributing nearly the same [27]. Focusing on vegetable crops alone, India is by far the largest single-country exporter to Kuwait, followed by the United States and Australia, demonstrating that Kuwait's food security strategy reaches far beyond the region and is truly global in scale [27].

### 1.4.3. Global Threats

This wide-reaching diversity of trading partners means that natural disasters, political upheaval, or disruptions in transportation services almost anywhere in the world threaten food security in Kuwait to some extent. The COVID-19 pandemic illustrates the consequences of this, as the breakdown of supply chains resulted in Kuwait's now infamous "onion crisis" of 2020 and other subsequent shortages of agricultural products imported from abroad, forcing authorities to implement rationing and exercise direct control over the distribution of scarce supplies while issuing pleas to farmers to shift production toward these previously imported crops [28,29]. This represents a strong motivator for the country to start looking to new, innovative systems and technology that can add to its traditional strategies toward greater food sovereignty.

### 1.5. Solutions from Advances in Controlled Environment Agriculture

The twin challenges of minimal arable land and extreme water stress that face Kuwait and other countries in the GCC make it imperative to develop and adopt sustainable crop production systems that contribute to the enhancement of agricultural productivity in an efficient and context-specific way. Advanced CEA systems configured as indoor farms or vertical farms may offer a possible solution. Indoor farming produces crops within a structure that provides protection from the external environment and control over growing conditions. These farms are typically soilless, using chemically inert substrates to anchor crops, offering a high level of control over the application of water and nutrients while avoiding issues with soil borne pathogens, pests and contaminants [30,31]. Irrigation is accomplished through hydroponic or aeroponic systems, which are known for their high water use efficiency, regularly demonstrating over 90% reduction in water usage compared to traditional field agriculture [30,32–39]. To further increase resource-use efficiency while diversifying nutritional yield, indoor farms can also make use of aquaponic systems, where fish or crustaceans are cultivated in tanks to produce animal protein while naturally enriching the water used for the irrigation of plant-based commodities [11,32,36,40]. In all of these systems, traditional fertilizer is replaced by carefully calibrated nutrient solutions customized for optimal plant health, eliminating wasteful over-application and nutrient diversion through soil adsorption or leaching that occurs in traditional soil-based cultivation [30,37]. Indoor farming systems are typically configured in a single horizontal layer similar to conventional agriculture, requiring a large building footprint for commercial production. Alternately, these systems may be oriented in vertically stacked arrays, requiring a different structural support system but offering the benefit of a much smaller building footprint [30,37]. Farming upward—or "vertical farming"—is well-suited for urban applications where it can make use of existing structures and established municipal services to produce food close to the point of consumption [30,32,33]. Given Kuwait's

urban landscape and demographic, this last point makes vertical farming an especially attractive possibility, and one that may complement the existing urban design features of co-op market distribution and public lands interspersed within and around urban population centers.

## 2. Materials and Methods

In this study, statistical data on open and protected field area planted, quantity produced, and the actual crop net imports was obtained from the State of Kuwait Public Authority for Agriculture Affairs and Fish Resources (PAAFR) [10]. Data for the impact of advanced CEA technology on crop yields in indoor and vertical farms was obtained from a previous study [38]. It suggests that the addition of advanced CEA technology in an indoor, climate-controlled setting, and vertical stacking of crops would boost yields over open field or protected field cultivation.

The following steps were undertaken to quantify the potential for indoor and vertical farming to improve Kuwait's ability to reach national self-sufficiency for target vegetable crops domestically:

1.  To quantify the effect of currently employed agricultural techniques, the change in yield between target crops grown in open field conditions and protected field conditions in Kuwait was calculated. First, actual quantity produced (kg) was divided by actual area planted (m$^2$) to determine actual yield (kg/m$^2$) for each cultivation method. Then, the difference between protected field yield and open field yield was divided by open field yield and multiplied by 100 to achieve the actual percentage increase of protected field cultivation over open field cultivation.
2.  To estimate the potential increase in yield over open field cultivation for indoor farming and vertical farming, the actual open field yield was multiplied by the factor of yield increase due to the addition of advanced CEA technology for each target crop, as determined by Banerjee and Adenaeuer [38].
3.  To calculate the total area required to achieve 100% self-sufficiency of the target crops, the actual quantity (kg) of net imports for each crop was divided by the calculated yield (kg/m$^2$) for each of the four cultivation methods. The area required for the six target crops (m$^2$) was totaled according to cultivation method to achieve total area required. The totals were converted to square kilometers for ease of comprehension.

Target vegetable crops were selected based on the following criteria: (a) crops with the potential to be successfully cultivated and commercially viable for an indoor or vertical farm setting; and (b) crops with consistent statistical data to document that they are staples of the country by annually ranking among the top 15 most-consumed vegetable crops in Kuwait that were domestically cultivated under soil-based field conditions from 2007–2017. Target crops include tomatoes (Solanum lycopersicum), potatoes (Solanum tuberosum), cucumbers (Cucumis sativus), green peppers (Capsicum annuum), carrots (Daucus carota), lettuce (Lactuca sativa), and cabbages (Brassica oleracea), selected according to the set criteria and are currently documented as grown for commercial purposes in Kuwait, as the agricultural example for the focus of this study.

Open field cultivation indicates that the crop was grown in open field soil with full exposure to Kuwait's natural climatic conditions. Protected field cultivation indicates that the crop was grown in field soil under glass, fiberglass, rockwool, or nylon shelter of some configuration. No soilless cultivation of vegetable crops had taken place in Kuwait according to the most recently available statistics published in 2018, with the agricultural sector still relying on soil-based cultivation [10].

## 3. Results

### 3.1. Calculation of Yields for Current Agricultural Techniques

Table 1 shows the difference in yield for crops grown under basic protective conditions compared to open field cultivation. The statistical data shows that even with minimal, low-tech intervention, there is a definitive increase in yield for all crops.

**Table 1.** Actual 2017 crop yield in open and protected conditions in Kuwait.

| | Open Field Area Planted (m$^2$) [1] | Open Field Quantity Produced (kg) [1] | Open Field Yield (kg/m$^2$) [2] | Protected Field Area Planted (m$^2$) [1] | Protected Field Quantity Produced (kg) [1] | Protected Field Yield (kg/m$^2$) [2] | Increased Yield (Protected vs. Open Field) [2] |
|---|---|---|---|---|---|---|---|
| Tomato | 1,404,000 | 9,126,000 | 6.5 | 6,328,000 | 94,920,000 | 15.0 | 130.8% |
| Potato | 8,548,000 | 51,288,000 | 6.0 | 69,000 | 483,000 | 7.0 | 16.7% |
| Cucumber | 19,000 | 76,000 | 4.0 | 5,814,000 | 69,768,000 | 12.0 | 200.0% |
| Green Pepper | 100,000 | 351,000 | 3.5 | 2,493,000 | 13,712,000 | 5.5 | 56.7% |
| Carrot | 94,000 | 236,000 | 2.5 | 20,000 | 60,000 | 3.0 | 19.5% |
| Lettuce | 2,604,000 | 11,718,000 | 4.5 | 276,000 | 1,450,000 | 5.3 | 16.8% |
| Cabbage | 1,927,000 | 10,600,000 | 5.5 | 61,000 | 397,000 | 6.5 | 18.3% |

[1] Source data: [10]. [2] Calculated from source data: [10].

Using official government-reported statistical quantification of soil-based agricultural crop yields, Table 1 shows that protected field agriculture outperforms open field agriculture practices when it comes to yield increase for every target crop in this study. The increase in productive capacity for protected crops is demonstrated by greater weight harvested per unit area of land than their open field cultivated counterparts. This allows farmers to achieve an equivalent harvest from a smaller land area when planted under protection. This is evident when considering the observed production increase above 50% for tomatoes (130.8%), cucumbers (200%), and green peppers (56.7%) and even the modest yet noteworthy production increase above 15% for potatoes (16.7%), carrots (19.5%), lettuce (16.8%), and cabbage (18.3%). This demonstrates that even the basic forms of agricultural field modification (providing protective shading and greenhouses) can have a noticeable effect on crop yield production. Tomatoes grown under protective field conditions yield an additional 8.5 kg/m$^2$ over tomatoes grown in open field conditions. Protective conditions increase potato yield by 1 kg/m$^2$ over open conditions. Cucumber gains 8 kg/m$^2$, green pepper gains 2 kg/m$^2$, carrot gains 0.5 kg/m$^2$, lettuce gains 0.8 kg/m$^2$ and cabbage gains 1 kg/m$^2$ over open field yields.

Three target crops—tomato, cucumber, and green pepper—achieved an increase of greater than 50% from basic protective measures. The data show that farmers have already adjusted their planting scheme accordingly, placing over 90% of each of these crops under protection. It is noteworthy that these three crops also have the highest rates of self-sufficiency, at 64%, 99%, and 42%, respectively [10]. Cucumber can be grown so effectively through protected soil-based cultivation that self-sufficiency has already been reached for this crop, and so cucumber shall be excluded from the calculations below.

*3.2. Calculating Potential Yield Increase with CEA*

A study by Banerjee and Adenaeuer suggests that the addition of advanced CEA technology in an indoor, climate-controlled setting would boost yields even further, increasing tomato by a factor of 3.4, potato by 5.4, green pepper by 4.4, carrot by 1.9, lettuce by 1.5, and cabbage by 1.3 [38]. Vertical stacking of these crops using the same advanced CEA technology could result in yields increased by a factor of 548, 552, 704, 347, 709, and 215, respectively. To reach these figures, Banerjee and Adenaeuer studied a model where the crops were configured in a vertical array across multiple floors of a high-rise building using a staggered planting scheme to facilitate nearly continuous year-round harvest [38]. Table 2 shows the potential yield increase of applying factors of advanced CEA technology to actual open field yield of these crops in Kuwait.

**Table 2.** Potential Yield Increase with Indoor and Vertical Farming in Kuwait. [1]

|  | Actual Yield Open Field (kg/m$^2$) | Actual Yield Protected Field (kg/m$^2$) | Potential Yield Indoor Farm (kg/m$^2$) | Potential Yield Vertical Farm (kg/m$^2$) |
|---|---|---|---|---|
| Tomato | 6.50 | 15.00 | 22.10 | 3562 |
| Potato | 6.00 | 7.00 | 32.40 | 3312 |
| Green Pepper | 3.51 | 5.50 | 15.40 | 2471 |
| Carrot | 2.51 | 3.00 | 4.80 | 871 |
| Lettuce | 4.50 | 5.25 | 6.75 | 3191 |
| Cabbage | 5.50 | 6.51 | 7.15 | 1183 |

[1] Calculated from source data: [10].

Table 2 paints a clear picture of how the demonstrated yield improvements currently employed by Kuwaiti farmers (shown as "Actual Yield Protected Field") can be further improved upon through application of advanced CEA technology in the form of either indoor farms or vertical farms. Applying the factors of increase determined by Banerjee and Adenaeuer to the actual open field yield, this table shows the potential increased yield if the same amount of land area were used to plant the target crop using indoor farm technology or vertical farm technology, respectively [38]. In an indoor farm, tomato would have the potential of yielding 22.1 kg/m$^2$, whereas in open field cultivation in Kuwait, tomato crops yield only 6.5 kg/m$^2$, for a projected increase of 240%. Likewise, potential potato yield of 32.4 kg/m$^2$ would be an increase of 440%, potential green pepper yield of 15.4 kg/m$^2$ would be an increase of 340%, carrot yield would increase 92% to 4.8 kg/m$^2$, lettuce yield would increase 51% to 6.8 kg/m$^2$ and cabbage yield would increase 31% to 7.2 kg/m$^2$. Repeating the calculation using Banerjee and Adenaeuer's factors of increase for vertical farming reveals that a tomato yield of 6.5 kg/m$^2$ in open field would have the potential of being 3562 kg/m$^2$ had it been produced in a multi-level vertical farm, a projected (54,700%) increase [38]. The projected potato yield of 3312 kg/m$^2$ would be an increase of 55,100%, green pepper would increase 70,300% to 2464 kg/m$^2$, carrot would increase 34,620% to 868 kg/m$^2$, lettuce would increase 70,811% to 6.8 kg/m$^2$ and cabbage would increase 21,409% to 7.2 kg/m$^2$. These figures clearly show the succession of actual yield increase from open field cultivation to protected field cultivation, and then the projected yield increase over open field cultivation for indoor farm cultivation and finally for vertical farm cultivation.

In actual practice, yield increase in indoor or vertical farms in Kuwait may far exceed these estimates, given that Banerjee and Adenaeuer based their calculations on farm data from Germany, where a more temperate climate results in greater baseline yield for field-grown crops [38]. The idea that the increase in yield could be drastically greater in Kuwait is backed up by a study by Barbosa et al. conducted in a region of Arizona, USA, with similar climatological conditions to Kuwait [39]. The study compared lettuce crop yield in outdoor soil-based conditions to soilless hydroponic cultivation in controlled indoor conditions, finding that the latter resulted in 11 ($\pm$1.7) times greater yield per area than the former. This physical demonstration in a desert climate far outstrips Banerjee and Adenaeuer's estimate of a 1.5 times increase for lettuce crops based on farm data collected in a temperate climate. It is conceivable that similar outcomes could be achieved for other crops.

### 3.3. Projected Indoor and Vertical Farm Area Required for Self-Sufficiency

In 2017, PAAFR reports the actual net import weight of the six target crops as follows: tomato, 57,473 $\times$ 10$^3$ kg; potato, 92,358 $\times$ 10$^3$ kg; green pepper, 19,371 $\times$ 10$^3$ kg; carrot, 20,617 $\times$ 10$^3$ kg; lettuce, 17,178 $\times$ 10$^3$ kg; and cabbage, 10,051 $\times$ 10$^3$ kg [10]. Dividing the actual net import weight of the target crops by the calculated yield under each planting scenario in Table 2 results in the total estimated planting area for each target crop required to achieve 100% self-sufficiency, as shown in Table 3.

**Table 3.** Estimated area (square meters) needed to reach 100% self-sufficiency by cultivation method [1].

|  | **Open Field** | **Protected Field** | **Indoor Farm** | **Vertical Farm** |
| --- | --- | --- | --- | --- |
| Tomato | 8,842,000 | 3,831,533 | 2,600,588 | 16,135.04 |
| Potato | 15,393,000 | 13,194,000 | 2,850,556 | 27,885.87 |
| Green Pepper | 5,534,571 | 3,522,000 | 1,257,857 | 7839.34 |
| Carrot | 8,246,800 | 6,872,333 | 4,481,957 | 23,670.49 |
| Lettuce | 3,817,333 | 3,241,132 | 2,544,889 | 5383.27 |
| Cabbage | 1,827,455 | 1,546,307 | 1,405,734 | 8496.20 |

[1] Calculated from source data: [10].

Adding the area required for the six target crops under each cultivation scenario results in an estimation of the total area required if 100% self-sufficiency were to be achieved entirely through use of each individual method, as shown in Figure 1.

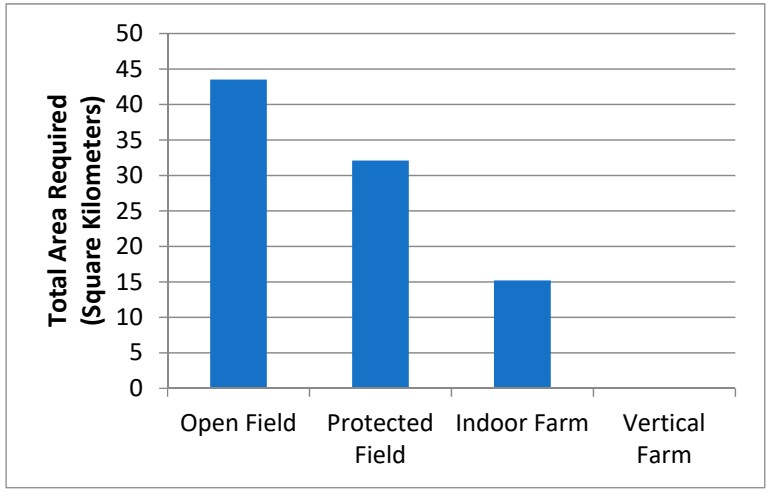

**Figure 1.** Comparison of total land area required to achieve self-sufficiency using different cultivation scenarios (calculated from [10]).

Consumption of target vegetable crops exceeds current agricultural output in Kuwait, with the country importing the remainder to ensure that demand is met. It is important to state that this study does not suggest that traditional agriculture should be replaced, but that new techniques and technologies may help the country supplement the output of traditional farmers to improve national self-sufficiency by producing more food within its borders. For this reason, calculations in Table 3 are based on import data alone rather than total consumption, as the goal is to reduce imports.

To determine how much land area would need to be devoted to growing the target crops in order to match the current production deficit, the actual quantity of imports is divided by the actual/projected yield of the four cultivation methods considered in this study, resulting in the estimated area in square meters shown in Table 3. Converting the figures for open field cultivation to square kilometers for ease of comprehension results in a planting area of 8.8 km$^2$ for tomato, 15.4 km$^2$ for potato, 5.5 km$^2$ for green pepper, 8.2 km$^2$ for carrot, 3.8 km$^2$ for lettuce, and 1.8 km$^2$ for cabbage, for a grand total of 43.5 km$^2$ (see Figure 1). Thus, in order to achieve 100% self-sufficiency for these crops, the country would have to expand open field agriculture by 159% over the current actual planted open field area of 16.8 km$^2$ for these six crops. This would equate to 36% of all arable land currently used for cultivating all vegetable crops (121 km$^2$). Considering the overall scarcity of arable land and competition for use (i.e., pasture), the need for full irrigation, and the lack of freshwater resources (see Section 1.2), expanding open field cultivation to accommodate this increase is not a viable option. Due to its increased yield, protected field agriculture would reduce the total planting area required to 32.1 km$^2$. Although this is a

reduction of 26% over the total open field requirement, protected field agriculture is still subject to the same environmental resource limitations as open field farming. While some of these challenges are being addressed through the scientific development of aquaculture, Desert Agriculture and Ecosystems Programs and other biotechnology, environmental, and agricultural programs at the Kuwait Institute for Scientific Research, the fact remains that limited arable land and natural resources will continue to be a major challenge for the agricultural sector in Kuwait [2,41]. With this in mind, Table 3 shows that the indoor farming scenario would require planting of only 15.2 km$^2$, cutting down on the land area required by 65% over open field farming, whereas vertical farming according to the cited model would require less than 0.1 km$^2$ (approximately 0.089 km$^2$), a 99.8% reduction in the area required by open field farming for the six target crops. Advanced CEA systems utilized in indoor and vertical farming also have the crucial benefit of employing soilless culture, thus bypassing the need for occupying Kuwait's scarce arable land, as they can be sited on wasteland or any surface suitable for construction. They also utilize extremely resource-efficient water and nutrient delivery systems, alleviating some concern over increasing water stress and soil salinity in the natural environment. Because the estimated potential indoor and vertical farm yields in Table 2 may be drastically low, the calculations of required area shown in Table 3 may, in turn, be greater than actual need, indicating that improved self-sufficiency for vegetable crops may be within closer reach than even these calculations suggest.

## 4. Discussion

### 4.1. Implications of Findings

Taken together, these findings indicate that indoor farming and vertical farming merit serious consideration and study of the feasibility of investing in development and adoption of these methods to supplement crop yields attained by traditional farming for a more resilient food security. Traditional farming and advanced CEA could work in tandem to achieve maximum environmental, social, and economic benefits. Shifting more sensitive crops to indoor or vertical farms and focusing conventional soil-based agriculture on those crops best suited for the arid climate could increase resource use efficiency and encourage better environmental outcomes, such as reducing the over-extraction of groundwater for irrigation and slowing the rate of soil salinization while maintaining farmer livelihoods and preserving—and perhaps reviving—traditional knowledge of desert agriculture. At the same time, advanced CEA technologies may allow Kuwait to begin producing new types of crops that are not possible with conventional agriculture in a desert climate, even in protected field conditions. In particular, plants of specialized culinary or medicinal value could be locally produced and find a strong market in the metropolitan society of Kuwait and other GCC countries.

It is also important to recognize that there are more crops of importance beyond those featured in this paper that should be incorporated into a well-rounded indoor or vertical crop production plan. It is unlikely that 100% self-sufficiency would be reached for every possible crop. However, these calculations show just how little land area could be required to substantially improve the self-sufficiency of vegetable production in Kuwait and similar GCC countries by supplementing traditional agriculture with indoor and vertical farming. By analyzing the import sources and potential risk of supply disruption with consideration for the relative importance of the crop for Kuwaiti dietary preference and well-being, a balanced production strategy could be developed that would guide the design and implementation of indoor agriculture projects, as well as determine permitting quotas and funding incentives.

### 4.2. Challenges of Vertical Farming in Kuwait and Other GCC Countries

Although the potential for higher crop yields is clear and compelling, initiating indoor and vertical farming is not without challenges. Regardless of geography, high startup costs and energy-intense operation are universally cited as drawbacks to initiating this new mode

of agriculture [30,33,35,38,39]. But these, along with the additional water consumption inherent with any increase in agriculture, take on a new dimension in the context of Kuwait and other GCC countries where water production and energy generation are coupled through the process of multi-stage flash desalination.

### 4.2.1. Coupled Water and Energy Production

With no natural freshwater resources to draw from, Kuwait relies on desalination to satisfy the growing demand for potable water [2]. The dominant process in the region is multi-stage flash desalination whereby fossil fuel is burned to pressurize salt water, producing steam that runs energy generating turbines before being condensed and distilled into fresh water [5,12,15]. This coupling of energy and water production means that any increase in demand for water requires additional energy production by default—and vice versa. While Kuwait and other GCC countries have no shortage of the primary fossil fuel and saline water inputs, the process is still costly both in economic and environmental terms. Foremost, the reality of climate change and the international commitments to reduce greenhouse gas emissions prevent indiscriminate increases in fossil fuel-based processes. Already, the largest share (58%) of greenhouse gas emissions in the energy sector originate from the production of electricity and desalinated water, leading Kuwait to look for ways to reduce these activities rather than increase them [5]. Additionally, disposal of brine—a toxic byproduct of the desalination process—poses further concern that must be considered before boosting production at these facilities [12].

### Minimizing Water Use

Implementing these farms would increase water demand despite the high-water efficiency rate of advanced CEA systems. To mitigate this effect, CEA could focus on those crops that require the greatest inputs when grown conventionally, freeing up Kuwait's limited arable land for crops that are best suited for outdoor cultivation and are better adapted to the arid climate. The highly efficient water and nutrient delivery system of CEA has the potential to support and maximize production of more sensitive crops while reducing the inputs that would otherwise be required if they continued to be grown conventionally [30,32–39]. Research and development into cultivars that are optimized for growth in each environment can further increase resource efficiency of crop production. Furthermore, integrating water recycling technology and gray water use into the design of these new facilities could be standard procedure as the sector develops.

### Minimizing Energy Use

Similar measures exist to help mitigate potential negative effects of increased energy demand. Indoor and vertical farming technologies are sophisticated and cutting edge, constantly being improved to optimize performance and minimize waste. Standardizing the integration of highly energy efficient equipment and renewable energy generation into the design of new farms could be considered [30,42,43]. Perhaps even more importantly, investment in renewable energy technology to partially replace fossil fuels at desalination facilities would reduce greenhouse gas emissions across all sectors [4]. Public outreach and education initiatives, as well as a strategic push for society-wide adoption of water and energy conservation best practices could also be part of a multi-pronged strategy to maintain or lower current production levels at desalination facilities as mitigation for the greater share of water and energy that indoor/vertical farming will require.

### 4.2.2. High Start-Up Costs

High start-up costs can pose a barrier to initiating advanced CEA systems. Compared to traditional agriculture which requires little infrastructure before planting can commence, setting up an indoor farm requires substantial investment in facilities and equipment at the outset. While some cost may be saved by retrofitting or rehabilitating an existing structure, systems for lighting, irrigation, and climate control must still be purchased and

installed, as well as substrates, nutrients and water provided to create a suitable growing environment for the intended crop [32]. However, where substantial barriers to traditional agriculture exist in the natural environment, such as in Kuwait and other GCC countries, the value of improved food sovereignty in the longer-term may outweigh the higher initial investment [38]. GCC countries are amongst the best suited nations to lead and develop innovative initiatives in the modern CEA industry due to their ability to bear the cost and reliably provide the required energy. However, this must be initiated with careful consideration and accounting for the environmental burden some of the technologies incur [33,44].

Subsidizing Indoor and Vertical Farm Start-Ups

Government intervention in the form of subsidies, insurance, co-ownership or other forms of backing would lend security to farmers hesitant to accept the risk of adopting new production technologies. Further, it could prevent the documented issue of devoting indoor farm production solely to high-profit crops, such as fast-growing leafy greens [30,37]. Support from the government and its national or international non-governmental and private sector partners could promote planting schemes that target varieties of produce necessary to address the country's critical self-sufficiency deficits, while expanding or spearheading research into improving cultivars optimized for indoor/vertical farming applications. Strategic funding would serve to stimulate and diversify the economy, as this technology has the potential to bring with it new employment and research opportunities from different backgrounds and disciplines (such as engineering and construction, biochemistry and biotechnology, skilled maintenance and other new service based industries) [1,30,32,37]. Aspects of indoor and vertical farms located in urban areas may also offer educational and development possibilities that could benefit the economy, improve social cohesion, and revitalize urban neighborhoods and real-estate values [32,44,45]. The current program of agricultural subsidies and food price stabilization demonstrates the willingness of Kuwait's government to invest not only for the benefit of the agricultural sector, but also for the quality of life for consumers and the overall standard of living for all within its borders [5]. Allocating support for indoor and vertical farming initiatives falls in line with these same goals.

*4.3. Institutional Mobilization and Policy Alignment*

Successfully expanding the agriculture sector to include indoor and vertical farming requires attention and coordination at the institutional level. Clear legislation designating oversight roles and quality control standards, as well as regulations regarding permitting, zoning, access to public services, and funding will create an enabling policy environment for farmers and entrepreneurs to quickly transition into setting up and operating indoor farms [3,38,44]. Although a complex and multi-institutional undertaking, it is not without precedent in Kuwait and, in fact, aligns with the government's stated vision and development strategy for the country.

Since 2010, Kuwait's development trajectory has been guided by "Kuwait Vision 2035", a series of five five-year plans for strategic national development accompanied by a framework for implementation and monitoring called the Kuwait National Development Plan (KNDP). After Kuwait officially adopted the United Nations' 2030 Agenda for Sustainable Development in 2015, its international commitments were fully integrated into the ongoing work of the KNDP, bringing together over 85 competent authorities and national partners to strive toward a common set of goals [9,46]. These efforts demonstrate that mass mobilization for sustainable development is already underway in Kuwait and can be harnessed to advance the adoption of CEA technology and indoor/vertical farming in particular.

Pillars of the Kuwait Vision 2035, such as effective public administration, diversified economy, sustainable living environment and creative human capital harmonize with the 2030 Agenda's Sustainable Development Goals (SDGs), including sustainable production

and consumption, sustainable cities, resilient infrastructure, and food security, improved nutrition, and promotion of sustainable agriculture [9]. These same programs emphasize "the critical role of scientific research in economic development and overall sustained peace and prosperity" [9] (p. 17). CEA, including aquaculture, indoor farming, and vertical farming in urban areas, and the suite of benefits previously described integrate seamlessly into the development apparatus already in motion in Kuwait. Furthermore, CEA projects can be viewed as a bridging initiative between the government and private sector stakeholders who spearhead cutting edge projects and push the envelope on innovation, solutions, and financing for development and growth. By embracing and actively supporting CEA projects, the government can be the catalyst for accelerated innovation, collaboration across business sectors, and among business, government, academia, and civil society through the implementation of sustainable solutions.

## 5. Conclusions

The artificial, optimized growing conditions of indoor farming systems leads to higher, more consistent yields while offering the option of expanding production to crop varieties that cannot be grown efficiently outdoors in an arid region. Such technologies come with their own unique set of challenges, whether technical, economic, or environmental, which can be further complicated by factors specific to a particular context, such as Kuwait's interlinked water-energy production system. Yet with smart design, innovative development, and a supporting policy framework, CEA offers the potential for fresh food production in countries with environmental limitations coupled with local and regional food security concerns.

Using advanced CEA technology in an indoor, climate-controlled setting has the demonstrated potential to multiply yields of important vegetable crops grown in a desert climate by 11 times or more. Configuring these indoor farms as a vertical array further increases the potential yield without the need for sprawling construction projects. These can be sited within urban centers to make fresh, nutritious food available without the need for long-haul transportation and storage. Building approximately 15 km$^2$ of indoor farms could entirely eliminate the need to import six important food crops (tomato, potato, green pepper, carrot, lettuce, and cabbage). It would require less than 0.1 km$^2$ to achieve the same using multi-level vertical farming configurations. Although a more balanced and diversified range of crops would be desirable in reality, these calculations reveal the potential for advanced CEA technologies to put greater food self-sufficiency within reach of GCC countries, reducing dependence on imported foods and increasing resilience to food supply disruption amid disaster, conflict, or other crises. This safety net offers opportunities for economic development through diverse job creation, enhanced private-public collaboration and multi-disciplinary research and development initiatives that foster innovation and pioneer new strategies for resource use efficiency not only in food production, but across society.

For indoor agriculture to realize the full potential benefits in a severely water-stressed region with complex socio-political dynamics, interventions must be planned for and comprehensively crafted in a highly contextual manner. Indoor and vertical agriculture is a relatively a new concept in the GCC and no major government projects have yet been built. GCC countries have a unique opportunity to learn from the experience of other countries that have successfully adopted and implemented CEA systems, allowing them to contextualize the way such systems could be integrated into GCC nations in a way to best meet their declared national and international development goals.

**Author Contributions:** Conceptualization, methodology, by M.J.A.; validation, Z.Z.; formal analysis, investigation, resources, data curation, writing—original draft preparation, M.J.A.; writing—review and editing, M.J.A. and Z.Z.; supervision, funding acquisition, K.M. All authors have read and agreed to the published version of the manuscript.

**Funding:** This work was supported in part by JSPS KAKENHI Grant Number JP17H00794.

**Data Availability Statement:** Publicly available datasets were analyzed in this study. This data can be found here: [https://www.csb.gov.kw/pages/Statistics_en?ID=18&ParentCatID=2, accessed on 30 January 2021].

**Conflicts of Interest:** The authors declare no conflict of interest. The funders had no role in the design of the study; in the collection, analyses, or interpretation of data; in the writing of the manuscript, or in the decision to publish the results.

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
