# Peer review of "Potential for Food Self-Sufficiency Improvements through Indoor and Vertical Farming in the Gulf Cooperation Council: Challenges and Opportunities from the Case of Kuwait"

_sustainability, doi:10.3390/su132212553_

Round 1

Reviewer 1 Report

The manuscript is on the future of indoor and vertical farming in urban environments. With overpopulation and climate change, the issue is more and more relevant.

The Introduction is well-written and useful.

Line 54: self-sufficiency can be calculated by several ways. On the long term, there are factors that may not be easy to ensure in this artificial context. For example, lacking genetic mixing and variability is a shortcoming, leading to inbreeding and genetic depression. Shorter-term needs (e.g. nutrients, water) are easier to provide, for sure.

Many crop species depend on pollinators, what are the possible solutions for this problem? You may consider that the functioning of pollinating insects highly depends on land use types:

https://www.sciencedirect.com/science/article/pii/S1470160X18307441?casa_token=ia03nLQ__sYAAAAA:jfvU8GM3UBu1cZsOOHgYmro0V6XL2ymbfxOaslXXpwLRGbvloe0sIJYRMpxxoaBIclUtfFbWXbqv

1.1 – 1.3 are nice presentations of the context

1.4 is clearly very important but my feeling is that it is somewhat out of the scope of this research. You may consider deleting or shortening this part of the manuscript.

Lines 206-213: these are very clear objects. You may consider that these are mostly short-term visions and you may briefly mention longer-term issues (genetics, sustainability, ecological stability, invasive pests).

The numbers in Table 2 are very convincing about vertical farming. To provide a complete picture, you may also mention disadvantages (even if the sum is still very positive). For example, it seems to be more expensive to irrigate vertical farms. Also, maintaining vertical farms may have a negative effect on the condition of buildings (given water is the most dangerous chemical for any instruments, on the long term)?

In urban environments, simple ecological communities can be found but also they have some diversity with inter-specific interactions among species. For vertical farming solutions, basic processes in urban ecological communities can be considered, for example, the presence of epiphytes:

https://link.springer.com/article/10.1007/s42974-020-00001-y

or decomposers:

https://link.springer.com/article/10.1007/s42974-021-00043-w

The paper is very well written, clear and well-structured. Scientifically speaking, it is mostly descriptive.

Author Response

Point 1: The manuscript is on the future of indoor and vertical farming in urban
environments. With overpopulation and climate change, the issue is more and more relevant.
Response 1: We appreciate the encouraging comment. We are glad that we have been able to
successfully express the relevance of our research to explore the potential for success of these
technologies in the GCC region.
Point 2: The Introduction is well-written and useful.
Response 2: Thank you. We worked hard to make it so and appreciate having the effort
recognized.
Point 3: Line 54: Self-sufficiency can be calculated by several ways. On the long term, there
are factors that may not be easy to ensure in this artificial context. For example, lacking
genetic mixing and variability is a shortcoming, leading to inbreeding and genetic depression.
Shorter-term needs (e.g. nutrients, water) are easier to provide, for sure. 

Response 3: Yes, the long-term implications of this technology are an important point that
needs to be considered. This paper champions the adoption of advanced CEA at the current
stage with the understanding that these are cutting edge technologies that are constantly
evolving. Implementation of commercial-sized CEA facilities at this stage allows the country
to benefit from short-term improvements in self-sufficiency while engaging in research and
development to pro-actively address emerging challenges, such as genetic variation, as you
mentioned. Section 4.3 and the Conclusion section of the manuscript promote the critical
need to strengthen research and development capacity in tandem with adoption of CEA
technology. [Lines 543-553, 576-580]
Point 4: Many crop species depend on pollinators, what are the possible solutions for this
problem? You may consider that the functioning of pollinating insects highly depends on
land use types [link to article].
Response 4: Thank you for directing us to the article, "The vulnerability of plant-pollinator
communities to honeybee decline: A comparative network analysis in different habitat types."
In response to your question related to our paper, it depends on the type of technology you
are adopting and the specific crop you want to produce. High-tech facilities (like many
vertical farms with artificial lighting) typically opt for mechanical methods to pollinate
fruiting crops, as a growing environment optimized for a plant may be harmful (or less
optimal) for a pollinating insect. Lower-tech facilities that do not require a sterilized
environment may opt for a more natural insect-based pollination method.
Point 5: 1.1-1.3 are nice presentations of the context.
Response 5: Thank you for your encouraging comment.
Point 6: 1.4 is clearly very important but my feeling is that it is somewhat out of the scope of
this research. You may consider deleting or shortening this part of the manuscript.

Response 6: As you can imagine, this is a highly complex and sensitive region which is
difficult to encapsulate, though we have striven to make this section as concise as possible.
While the main thesis of Section 1.4 is to point out the manifold reasons why the case study
country should consider new methods to enhance self-sufficiency, it also serves to make clear
that it is intimately connected within the region and beyond. This puts Kuwait in a position to
become an "influencer" or leader of innovation by championing advanced CEA technology as
a potential solution to food security threats in the region.
Point 7: Lines 206-213: these are very clear objects. You may consider that these are mostly
short-term visions and you may briefly mention longer-term issues (genetics, sustainability,
ecological stability, invasive pests).
Response 7: Research and development to address longer-term challenges to implementing
advanced CEA is promoted in Section 4.3 and the Conclusion section of the manuscript.
Please also see Response 1. [Lines 543-553, 576-580]
Point 8: The numbers in Table 2 are very convincing about vertical farming. To provide a
complete picture, you may also mention disadvantages (even if the sum is still very positive).
For example, it seems to be more expensive to irrigate vertical farms. Also, maintaining
vertical farms may have a negative effect on the condition of buildings (given water is the
most dangerous chemical for any instrument, on the long term)?
Response 8: There are a variety of challenges to implementing vertical farming, as you
mention. A plethora of published works on advanced CEA give a brief overview of the
disadvantages that are commonly encountered regardless of geographical location, and so, to
contribute new material to the current body of literature, the authors have chosen to focus on
the potential challenges and solutions that are unique to the context of the GCC region, which
is done in Section 4.2. [Lines 435-498]
Point 9: In urban environments, simple ecological communities can be found but also they
have some diversity with inter-specific interactions among species. For vertical farming
solutions, basic processes in urban ecological communities can be considered, for example,
the presence of epiphytes or decomposers [links to two articles].
Response 9: Thank you for directing us to the articles, "Could epiphytes be xenophobic?
Evaluating the use of native versus exotic phorophytes by the vascular epiphytic community
in an urban environment" and " The relationship between dung beetle diversity and manure
removal in forest and sheep grazed grasslands." You raise some interesting points for soil-
based urban agriculture. However, these considerations are beyond the scope of this paper
and deserve to be the focus of a separate study.
Point 10: The paper is very well written, clear and well-structured. Scientifically speaking, it
is mostly descriptive.
Response 10: Your recognition of our effort is highly appreciated and reassuring that the
manuscript is received as intended.

Reviewer 2 Report

I was delighted to review this paper, which I find highly relevant and well written.

Some minor suggestions for improvement are (line by line):

9-11. I suggest to avoid starting with a negative statement ("despite"). Please consider the following instead: 
"The countries in the Gulf Cooperation Council (GCC) are considered food secure due to their ability to import sufficient food to meet their population's demand, despite the considerable limitations to conventional agriculture".

13. Remove "Using" and begin the sentence with "Advanced".

17. Please specify which crops.

25. Please reduce the number of keywords to maximum 5.

53. Please specify the methods for data collection and analysis.

54. Please specify which crops.

71. It would be interesting to know the percentage of desert area surface within Kuwait's land mass.

99-109. Kuwait's population has doubled in the last 20 years. What is the impact of this population growth in food consumption patterns? It would be interesting to know this not only in terms of food demand, but also in terms of gastronomic traditions and the disappearance/substitution of endemic crops.

206. I could not find the equation, formula or method to perform these calculations (1, 2 and 3), the tables only refer to the source (the statistics available at the State of Kuwait Public Autority for Agriculture Affairs and Fish Resources). Please explain them in the methods.

215-220. I suggest to start section 2 with this paragraph.

224. Please indicate the parameters including the time-frame of the measures.

235. Year, month?

273. Excellent pacing and structure. Every time the data raised a question, I found it immediately answered.

338-341. This is a highly valuable clarification.

372. This study would be enriched with the calculation of water consumption of open agriculture vs vertical stacking systems.

397-398. I somewhat disagree, thus the suggestion to explore traditional gastronomic culture before population growth.

424. Are there any works to advance towards the use of renewable energies? I can imagine that solar and wind powered generators could be of great use in arid regions such as Kuwait.

435. Please argument the statement on water efficiency with calculations and/or literature.

474. Also donors such as FAO (who works under a silos system that includes capacity building, promoting policy changes and project implementation) or the WFP.

539. Specify.

Author Response

Response to Reviewer 2 Comments

Point 1: I was delighted to review this paper, which I find highly relevant and well written. Some minor suggestions for improvement are (line by line):

Response 1: We are happy that our paper met your approval and that you enjoyed engaging with it. Thank you for taking the time to suggest changes to improve the manuscript.

Point 2: 9-11. I suggest to avoid starting with a negative statement ("Despite"). Please consider the following instead: "The countries in the Gulf Cooperation Council (GCC) are considered food secure due to their ability to import sufficient food to meet their population's demand, despite the considerable limitations to conventional agriculture."

Response 2: Your suggestion is well-received and we have changed the manuscript accordingly. [Lines 9-12]

Point 3: 13. Remove "Using" and begin the sentence with "Advanced."

Response 3: Your suggestion is well-received and we have changed the manuscript accordingly. [Line 14]

Point 4: 17. Please specify which crops.

Response 4: We recognize the importance of specifying which crops, however adding the names of the six vegetables would exceed the 200 word limit for the abstract. Instead, we changed the phrase "food crops" to the more specific phrase "vegetable crops" to avoid confusion or misunderstanding regarding the nature of the agricultural products to which we are referring. [Line 18]

Point 5: 25. Please reduce the number of keywords to maximum 5.

Response 5: The number of keywords has been reduced to 5. [Lines 26-28]

Point 6: 53. Please specify the methods for data collection and analysis.

Response 6: The suggestion is well-received and has been addressed in the text and is further elaborated upon in the methodology section as requested in Point 10. [Lines 55-57, 221-240]

Point 7: 54. Please specify which crops.

Response 7: Your suggestion is well-received and we have changed the manuscript accordingly. [Line 58-59]

Point 8: 71. It would be interesting to know the percentage of desert area surface within Kuwait's land mass.

Response 8: With the exception of some small pockets of coastal wetlands along the Persian Gulf, Kuwait's entire surface is desert. The indicated sentence mentions the complete lack of renewable water resources (both surface and groundwater) within the country's borders, and the following sub-section (1.2.) mentions that only 9% of the land area is considered arable. The authors feel this is sufficient context to indicate how the natural environment is a limiting factor for conventional agriculture. [Line 76]    

Point 9: 99-109. Kuwait's population has doubled in the last 20 years. What is the impact of this population growth in food consumption patterns? It would be interesting to know this not only in terms of food demand, but also in terms of gastronomic traditions and the disappearance/substitution of endemic crops.

Response 9: We agree that this is an interesting line of questioning. However, the focus of this paper is the current state of agricultural production concerning staple crops that have been in high demand for the previous ten years. While CEA can absolutely be used to domestically produce crops that satisfy new gastronomic trends, the main focus at this stage (while the government has not yet adopted the technology) is to offer solutions that strengthen food self-sufficiency to ensure the meeting of basic needs. Naturally, as the population has increased dramatically, the quantity of food required to meet increased demand has also risen. Regardless of culinary tastes, the staple crops targeted by this paper will support an increase in domestic supply to prevent increased reliance on imports.

Anecdotally, within the author's lifetime, the variety and quality of produce has greatly declined. Now there is greater abundance but less variety, in part because global cultivation practices have homogenized commercial crops, but also because several of the countries that Kuwait used to import heavily from have been affected by prolonged conflict that limit the availability of different varieties sourced from small farmers that were prized by Kuwaiti consumers in the past.    

Point 10: 206. I could not find the equation, formula or method to perform these calculations (1, 2 and 3), the tables only refer to the source (the statistics available at the State of Kuwait Public Authority for Agriculture Affairs and Fish Resources). Please explain them in the methods.

Response 10: The suggestion is well-received. The method to perform the calculations has been elaborated in the text. [Lines 221-240]

Point 11: 215-220. I suggest to start section 2 with this paragraph.

Response 11: We agree with your suggestion and have changed the placement of the indicated paragraph. [Lines 211-217]

Point 12: 224. Please indicate the parameters including the time-frame of the measures.

Response 12: According to your suggestion, we have changed, "crops with consistent statistical data to officially document that they are staples of the country and are cultivated in a measurable capacity under specific agricultural conditions," to, "crops with consistent statistical data to document that they are staples of the country by annually ranking among the top 15 most-consumed vegetable crops in Kuwait that were domestically cultivated under soil-based field conditions from 2007-2017." [Lines 251-254]

Point 13: 235. Year, month?

Response 13: Added, "published in 2018" to the end of the sentence. The month of publication does not appear with the data and so cannot be added. [Line 265]

Point 14: 273. Excellent pacing and structure. Every time the data raised a question, I found it immediately answered.  

Response 14: Thank you for your encouraging assessment.

Point 15: 338-341. This is a highly valuable clarification.

Response 15: Thank you. We agree.

Point 16: 372. This study would be enriched with the calculation of water consumption of open agriculture vs vertical stacking systems.

Response 16: Necessary data to calculate potential water savings for the six target crops is not available. A general calculation of potential water savings for all field-grown crops if cultivated using CEA would be incongruous with the targeted nature of the yield and area calculations that form the core of this article. Please note that information regarding potential water efficiency improvements of using CEA was strengthened with additional literature as you suggested in Point 19. [Lines 190-192, 464-467]

Point 17: 397-398. I somewhat disagree, thus the suggestion to explore traditional gastronomic culture before population growth.

Response 17: The indicated sentence, "It is unlikely that 100% self-sufficiency would be reached for every possible crop," is a realistic assessment of any paradigm-shifting project or initiative. As previously noted in Point 15, our objective is not to suggest that traditional agriculture be replaced, but that it should be supplemented by and be used in tandem with CEA technology for a more well-rounded domestic vegetable supply.

Point 18: 424. Are there any works to advance towards the use of renewable energies? I can imagine that solar and wind powered generators could be of great use in arid regions such as Kuwait.

Response 18: Yes, part of the Kuwait National Development Plan is to increase renewable energy production through a number of projects across various industries. For the purpose of this paper, we suggest the integration of renewable energy generation into new CEA projects in sub-section 4.2.1.2. [Lines 474-478]    

Point 19: 435. Please augment the statement on water efficiency with calculations and/or literature.

Response 19: Your comment is well-received and literature citations have been added. Please also note that water use efficiency is discussed in sub-section 1.5. [Lines 190-192, 464-467]

Point 20: 474. Also donors such as FAO (who works under a silos system that includes capacity building, promoting policy changes and project implementation) or the WFP.

Response 20: Your comment is well-received and the indicated sentence has been amended to include, "Support from the government and its national or international non-governmental and private sector partners" to acknowledge the role of these organizations while maintaining emphasis on the need for the government to take a leading, pro-active role to ensure an enabling policy and regulatory environment for large-scale CEA initiatives to achieve success in the national context. [Lines 504-505]

Point 21: 539. Specify.

Response 21: The crops have been specified as suggested. [Lines 570-571]
